# Mediator or moderator? The role of obesity in the association between age at menarche and blood pressure in middle-aged and elderly Chinese: a population-based cross-sectional study

Lin Zhang ![ORCID] ,[1] Liu Yang,[1] Congzhi Wang,[1] Ting Yuan,[2] Dongmei Zhang,[3] Huanhuan Wei,[2] Jing Li,[4] Yunxiao Lei,[2,5] Lu Sun,[6] Xiaoping Li,[6] Ying Hua,[7] Hengying Che,[8] Yuanzhen Li[6]

For numbered affiliations see end of article.

**Correspondence to**
Hengying Che;
2722659079@qq.com

## ABSTRACT

**Objective** We investigated the moderation/mediation between the age of menarche and obesity parameters in predicting blood pressure (BP) in middle-aged and elderly Chinese.

**Design** Our study is a population-based cross-sectional study.

**Setting** Participants in this study came from the China Health and Retirement Longitudinal Study (CHARLS).

**Participants** The analytical sample included 4513 participants aged 45–96 years.

**Main outcome measurements** Data were selected from the CHARLS, a cross-sectional study. Between-group differences were evaluated using $\chi^2$, t-test and one-way analysis of variance. The trend of related variables by characteristics was also tested using contrast analysis, as appropriate. Then, correlations between characteristics, moderator, mediator, and independent and dependent variables were used by Spearman's correlation test and Pearson's correlation test. Finally, the mediation analysis was performed by model 4 in PROCESS V3.3 macro for SSPSS, and moderation analysis was used by model 1 for assessment. All covariates were adjusted in the moderation or mediation models.

**Results** In the correlation analysis, body mass index (BMI) and waist circle (WC) level were positively correlated with both systolic blood pressure (SBP) and diastolic blood pressure (DBP) in women (BMI and DBP: r=0.221, p<0.001; WC and DBP: r=0.183, p<0.001; BMI and SBP: r=0.129, p<0.001; WC and SBP: r=0.177, p<0.001). Age of menarche was negatively correlated with DBP (r=−0.060, p<0.001). However, the age of menarche was not significantly correlated with SBP (r=−0.014, p=0.335). In the moderator analysis, after controlling for the potential confounders, the interaction term of obesity parameters×age of menarche was not significant for predicting either DBP (BMI: B=0.0260, SE=0.0229, p=0.2556, 95% CI −0.0189 to 0.071; WC: B=0.0099, SE=0.0074, p=0.1833, 95% CI −0.0047 to 0.0244) or SBP (BMI: B=0.0091, SE=0.0504, p=0.8561, 95% CI −0.0897 to 0.108; WC: B=−0.0032, SE=0.0159, p=0.8427, 95% CI −0.0343 to 0.028). All correlations were significant correlation between age of

menarche, obesity parameters and BP except the path of the menarche age→SBP (with the addition of the BMI indicator: β=−0.0004, B=−0.0046, p=0.9797, 95% CI −0.3619 to 0.3526; with the addition of the WC indicator: β=0.0004, B=0.0044, p=0.9804, 95% CI −0.3439 to 0.3526) in crude model. In general, after controlling for potential confounders, BMI (DBP: β=−0.0471, B= −0.2682, p=0.0021, 95% CI −0.4388 to −0.0976; SBP: β=−0.0515, B=−0.6314, p<0.001, 95% CI −0.9889 to −0.2739) and WC (DBP: β=−0.0474, B= −0.2689, p<0.001, 95% CI −0.4395 to −0.0984; SBP: β=−0.0524, B=−0.6320, p<0.001, 95% CI −0.9832 to −0.2807) partly mediated the relationship between age of menarche and BP.

**Conclusions** The interaction term of obesity parameters×age of menarche was not significant for predicting either DBP or SBP in women. Moreover, obesity parameters partly mediated the relationship between the age of menarche and BP.

## STRENGTHS AND LIMITATIONS OF THIS STUDY

⇒ This is the first large population study to examine the moderation between the age of menarche and obesity parameters in predicting blood pressure (BP) in middle-aged and elderly Chinese, as well as the mediation effects of obesity parameters on the relationship between age of menarche and BP.

⇒ This study included a large sample of 4513 middle-aged and older Chinese. The results presented in this article could be explored further in prospective cohort studies.

⇒ The main limitations of our research are related to the cross-sectional study and the self-reported method used for the assessment of age at menarche and most related confounders.

⇒ BP in our study was measured at home by professionally trained volunteers, and the next step is the use of clinical BP and ambulatory BP.

## INTRODUCTION

Blood pressure (BP) is the force of circulating blood against the walls of the body's arteries, the major blood vessels in the body. Hypertension is diagnosed if BP readings are 140/90 mm Hg or above on two different days. As a chronic disease,[1] hypertension or elevated BP is a severe medical condition that significantly increases heart attack, stroke, blindness, kidney failure and other complications. Hypertension risk factors[2–6] include stress, salt consumption, harmful use of alcohol, low intake of fruits and vegetables, saturated fat and trans fats, being overweight or obese, tobacco use, low diet in vitamin D, lack of physical activity, family history, aged 65 years or over and other coexisting diseases. The WHO said[7] an estimated 1.13 billion people worldwide have hypertension, and most (two-thirds) are living in low-income and middle-income countries. It means that most people have high BP. Compared with the global average, people in low-income and middle-income countries may bear a disproportionate and heavier burden of the disease, owing to several factors such as the ongoing nutritional transition in dietary consumption and energy expenditure, increasing trends in sedentary lifestyle, other modifiable risk factors and inadequate healthcare systems.[8] As a developing country, China has shown a relatively high and stable prevalence of hypertension among adults, especially in middle-aged and older people. Mahajan et al[9] showed the prevalence of hypertension among middle-aged and elderly Chinese was approximately 41.81% in 2015. However, the rate control was still below 30%. Recently, hypertension control has become one of the critical public health interventions.[10] Therefore, it emphasises the promotion of public health strategy to prevent the significant hypertension risk factors and assure the fullest available accessibility and use of quality health control for people with risk factors or who develop subclinical or overt hypertension. These actions are integral to a comprehensive public health strategy for hypertension promotion and prevention.[11–13] Though there is still much uncertainty about the aetiology of hypertension, one of the strongest risk factors was overweight/obesity. Thus, increased body mass index (BMI) or centrally located body fat (especially waist circle (WC)) increases the risk of developing hypertension.

Several studies[14–18] have reported that early biological maturation has been associated with obesity and BP in adolescents. Werneck et al[14] showed that behavioural and hereditary variables are more related to BP in late-maturing adolescents in adolescents. Widén et al[15] found that earlier pubertal timing was associated with higher adult BMI and diastolic blood pressure (DBP) in both sexes. Mueller et al[16] showed that earlier menarche was associated with higher waist circumference and BMI measured. Dreyfus et al[17] found that earlier age at menarche was associated with higher mean BMI among African–American and white women. Cao and Cui[18] found that earlier age at menarche was associated with decreased DBP. However, it is not completely understood whether the changes in biological processes in the early years can affect cardiovascular outcomes across the life span. As important indicators of obesity, BMI and WC are directly associated with biological maturation (based on menarche age). Evidence from several studies[19–21] indicates that menarche is related to BMI in later life. Power et al[19] found that timing of puberty was strongly associated with BMI for the earlier maturers at ages 7–33 years. Adair and Gordon-Larsen[20] showed that overweight prevalence rates were significantly higher in early-maturing adolescents. Must et al[21] showed that early maturation in female adults overweight is mostly a result of the influence of elevated relative weight on early maturation. It has been speculated that the association between biological maturation and hypertension risk factors (based on menarche age) can be mediated or moderated by obesity parameters. Although speculative, the mediating or moderating role of obesity on the effect of early maturation (based on menarche age) on BP in adulthood has not been classified. However, this is especially important, given that the onset of biological maturation, especially the age of menarche, now occurs at a much earlier age than in previous studies. Moreover, fully understanding the mediating effect of obesity on the association between early maturation (based on menarche age) and BP in adulthood could help identify women's groups that are at risk (based on obesity parameters) and those that should be put forward, designed to place the many diverse conceptual and practice approaches and accomplishments in the early intervention. The moderating effect between early maturation (based on menarche age) and risk of hypertension (obesity parameters) on BP may also be fully considered.

To date, few studies on the mediating and moderating analysis between early maturation (based on menarche age) and risk of hypertension (obesity parameters) were conducted in individuals aged ≥45 years. Thus, this study aimed to determine their association with age of menarche, obesity parameters and other confounding factors using the China Health and Retirement Longitudinal Study (CHARLS) from participants aged ≥45 years in China.

## METHODS

### Study design and setting

Data from CHARLS wave 1 (2011) were used in our research. The CHARLS is an ongoing national longitudinal study administered by the National School for Development (China Centre for Economic Research)[22] from 2011.

### Individuals

At baseline,[23] 6883 women were recruited for a longitudinal study; 2370 individuals were excluded (ie, absence of metabolic measures, absence of medication history, having no BP, and/or using antihypertension drugs). Finally, 4513 individuals were included in the analyses.

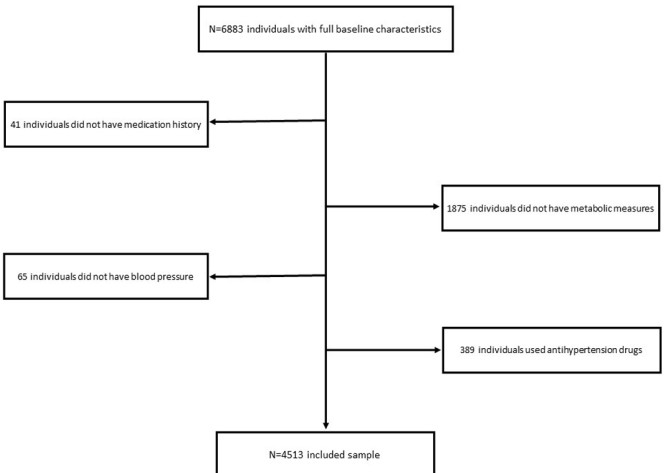

**Figure 1** Selection of participants.

Figure 1 summarises the selection of participants. Our study involved 4513 participants aged ≥45 years were total women (mean±SD age=58.59 ± 9.31 years, ranged from 45 years to 96 years). The mean and SD of menarche age were 16.27±2.18 years (ranged from 10 years to 30 years).

### Self-report of risk factors

We employed a general sociodemographic questionnaire that included the age, age of menarche (≤14, 15, 16, 17 and ≥18 years), educational attainment (illiterate, less than elementary school, high school, and above vocational school), marital status (single and married) and region (rural and urban). The lifestyle questionnaire included smoking status (never smoker, previous smoker and current smoker), alcohol consumption (never drinker, less than once a month and more than once a month), eating habits (≤2 meals per day, 3 meals per day and ≥4 meals per day), entertainment, the experience of a traumatic event (no and yes) and physical exercise habit (no, less than regular physical exercises and regular physical exercises). History of cardiovascular diseases and liver disease, antihyperlipidaemic drugs, and antidiabetic drugs were categorised in 'yes' or 'no'. Most variables were used in our previous studies.[24–30]

### Measurements

Triglyceride (TG) level, low-density lipoprotein cholesterol (LDL-C), high-density lipoprotein cholesterol (HDL-C), and glycosylated haemoglobin (HbA$_1$c) were analysed by the enzymatic colorimetric tests, serum creatinine (Scr) and serum uric acid (SUA) level were analysed using the urinalysis plus method. The average value of BP was measured by the mean of the three-time measurements using Omron HEM-7200 sphygmomanometer. Height was measured by the height measurement instrument. Participants were asked to stand barefoot on the base plate of the height metre in the upright position, with the back against the vertical backplate of the measuring instrument, with eyes open, arms alongside the body and heels were closed and the toes were separated by 60°. Anyone who appeared to unable to complete

measurement in time due to health reasons such as hunchback and inability to stand was excluded from our study. The method was used in our previous study.[30] BMI was calculated as body weight (kilogram) divided by the square of the subject's height (metre). WC was measured midway between the spina iliaca superior and the lower rib margin at the end of exhalation.

### Statistical analysis

The data were presented as means (M) and SD unless indicated otherwise. Mean and SD were used to describe continuous variables (age, age of menarche, LDL-C, HDL-C, TG, HbA$_1$c, Scr, SUA, BMI, WC, DBP and systolic blood pressure (SBP) level), and number and percentage were used to assess the categorical variables (educational attainment, marital status, region, alcohol consumption, smoking status, eating habits, entertainment, the experience of a traumatic event, physical activity, history of CVDs, history of liver diseases, history of antilipidaemic medication and antidiabetic medication). Between-group differences according to the age of menarche (≤14, 15, 16, 17 and ≥18 years) were evaluated by the analysis of variance (continuous data) or $\chi^2$ test (categorical data). Differences between the two categories of characteristics (marital status, region, entertainment, the experience of a traumatic event, history of CVDs, history of antilipidaemic medication, history of liver diseases and antidiabetic medication) were also evaluated using t-test. However, the trend of related variables (LDL-C, HDL-C, LDL-C/HDL-C, TG, HbA$_1$c, Scr, SUA, BMI, WC, and DBP and SBP levels) by above three categories of characteristics (age of menarche levels, educational attainment, alcohol consumption, smoking status, eating habits and physical activity) was also tested using contrast analysis. Correlations between characteristics and BP were used by Spearman's correlation test. However, relationships between continuous variables (age, LDL-C, HDL-C, LDL-C/HDL-C, TG, HbA$_1$c, Scr, SUA, BMI, WC and menarche age) and BP were used by Pearson's correlation test. The mediation analysis was performed by model 4 in PROCESS V.3.3,[31–33] and the moderation analysis was used by model 1 for assessment. All covariates (including age, educational attainment, marital status, region, smoking status, alcohol consumption, eating habits, entertainment, the experience of a traumatic event, physical activity, history of cardiovascular diseases, hepatitis history, antihyperlipidaemic drugs, antidiabetic drugs, LDL-C, HDL-C, TG, HbA$_1$c, Scr and SUA) were adjusted for in the mediation or moderation models. Analyses including the unstandardised coefficient (B), SE or standardised coefficient (β) with 95% CI were calculated using the setting of modelling building in IBM SPSS V.22.0 software for Windows V.10 using the bootstrap method with a 5% significance level.

### Patient and public involvement

No patients were involved in the development of the question, design or data interpretation.

**Table 1** Characteristics by age at menarche (N=4513)

| Variables | ≤14 years N=946 | 15 years n=671 | 16 years n=930 | 17 years n=702 | ≥18 years n=1264 | P value |
|---|---|---|---|---|---|---|
| Age (years) | 56.88±9.38 | 57.49±9.34 | 58.41±9.42 | 58.6±9.12 | 60.57±8.91 | <0.001* |
| Educational attainment | | | | | | |
| Illiterate | 300 (31.71) | 233 (34.72) | 378 (40.65) | 314 (44.73) | 649 (51.34) | <0.001 |
| Less than elementary school | 527 (55.71) | 383 (57.08) | 486 (52.26) | 343 (48.86) | 572 (45.25) | |
| High school | 82 (8.67) | 42 (6.26) | 48 (5.16) | 34 (4.84) | 34 (2.69) | |
| Above vocational school | 37 (3.91) | 13 (1.94) | 18 (1.94) | 11 (1.57) | 9 (0.71) | |
| Marital status | | | | | | |
| Single | 137 (14.48) | 79 (11.77) | 137 (14.73) | 98 (13.96) | 207 (16.38) | 0.101 |
| Married | 809 (85.52) | 592 (88.23) | 793 (85.27) | 604 (86.04) | 1057 (83.62) | |
| Region | | | | | | |
| Rural | 519 (54.86) | 426 (63.49) | 577 (62.04) | 454 (64.67) | 885 (70.02) | <0.001 |
| Urban | 427 (45.14) | 245 (36.51) | 353 (37.96) | 248 (35.33) | 379 (29.98) | |
| Cigarette smoking | | | | | | |
| Never smoker | 883 (93.34) | 629 (93.74) | 846 (90.97) | 646 (92.02) | 1156 (91.46) | 0.017 |
| Previous smoker | 8 (0.85) | 11 (1.64) | 18 (1.94) | 13 (1.85) | 38 (3.01) | |
| Current smoker | 55 (5.81) | 31 (4.62) | 66 (7.1) | 43 (6.13) | 70 (5.54) | |
| Alcohol consumption | | | | | | |
| Never drinker | 827 (87.42) | 607 (90.46) | 809 (86.99) | 620 (88.32) | 1100 (87.03) | 0.520 |
| Less than once a month | 45 (4.76) | 26 (3.87) | 48 (5.16) | 37 (5.27) | 65 (5.14) | |
| More than once a month | 74 (7.82) | 38 (5.66) | 73 (7.85) | 45 (6.41) | 99 (7.83) | |
| Eating habits | | | | | | |
| ≤2 meals per day | 131 (13.85) | 99 (14.75) | 123 (13.23) | 82 (11.68) | 172 (13.61) | 0.706 |
| 3 meals per day | 804 (84.99) | 567 (84.5) | 794 (85.38) | 609 (86.75) | 1073 (84.89) | |
| ≥4 meals per day | 11 (1.16) | 5 (0.75) | 13 (1.40) | 11 (1.57) | 19 (1.50) | |
| Entertainment | | | | | | |
| No | 417 (44.08) | 335 (49.93) | 470 (50.54) | 372 (52.99) | 655 (51.82) | 0.002 |
| Yes | 529 (55.92) | 336 (50.07) | 460 (49.46) | 330 (47.01) | 609 (48.18) | |
| Experience of a traumatic event | | | | | | |
| No | 881 (93.13) | 630 (93.89) | 867 (93.23) | 652 (92.88) | 1169 (92.48) | 0.837 |
| Yes | 65 (6.87) | 41 (6.11) | 63 (6.77) | 50 (7.12) | 95 (7.52) | |
| Physical activity | | | | | | |
| No physical exercise | 587 (62.05) | 393 (58.57) | 556 (59.78) | 428 (60.97) | 771 (61.00) | 0.260 |
| Less than regular physical exercises | 178 (18.82) | 137 (20.42) | 171 (18.39) | 136 (19.37) | 271 (21.44) | |
| Regular physical exercises | 181 (19.13) | 141 (21.01) | 203 (21.83) | 138 (19.66) | 222 (17.56) | |
| History of CVDs | | | | | | |
| No | 810 (85.62) | 590 (87.93) | 780 (83.87) | 605 (86.18) | 1115 (88.21) | 0.033 |
| Yes | 136 (14.38) | 81 (12.07) | 150 (16.13) | 97 (13.82) | 149 (11.79) | |
| Hepatitis history | | | | | | |
| No | 910 (96.19) | 647 (96.42) | 885 (95.16) | 670 (95.44) | 1226 (96.99) | 0.205 |
| Yes | 36 (3.81) | 24 (3.58) | 45 (4.84) | 32 (4.56) | 38 (3.01) | |
| History of antilipidaemic medication | | | | | | |
| No | 884 (93.45) | 628 (93.59) | 884 (95.05) | 656 (93.45) | 1184 (93.67) | 0.566 |
| Yes | 62 (6.55) | 43 (6.41) | 46 (4.95) | 46 (6.55) | 80 (6.33) | |

Continued

**Table 1** Continued

| Variables | ≤14 years N=946 | 15 years n=671 | 16 years n=930 | 17 years n=702 | ≥18 years n=1264 | P value |
|---|---|---|---|---|---|---|
| History of antidiabetic medication | | | | | | |
| No | 910 (96.19) | 645 (96.13) | 883 (94.95) | 663 (94.44) | 1213 (95.97) | 0.309 |
| Yes | 36 (3.81) | 26 (3.87) | 47 (5.05) | 39 (5.56) | 51 (4.03) | |
| LDL-C (mg/dL) | 119.66±35.04 | 118.04±35.61 | 120.62±34.81 | 121.79±36.79 | 121.45±35.27 | 0.065* |
| HDL-C (mg/dL) | 50.57±14.17 | 51.31±14.01 | 50.84±14.43 | 51.05±14.18 | 53.2±14.87 | <0.001* |
| LDL-C/HDL-C | 2.50±0.90 | 2.42±0.89 | 2.52±0.91 | 2.51±0.92 | 2.43±0.92 | 0.242* |
| TG (mg/dL) | 143.09±99.89 | 142.86±103.67 | 142.07±98.31 | 145.85±132.3 | 124.16±76.25 | <0.001* |
| HbA$_1$c (%) | 5.31±0.83 | 5.25±0.86 | 5.31±0.89 | 5.32±0.84 | 5.27±0.76 | 0.594* |
| Scr (mg/dL) | 0.69±0.14 | 0.69±0.14 | 0.70±0.18 | 0.68±0.13 | 0.69±0.15 | 0.813* |
| SUA (mg/dL) | 4.09±1.10 | 3.99±1.09 | 4.04±1.07 | 4.01±1.01 | 3.97±1.05 | 0.021* |
| BMI (kg/m$^2$) | 24.63±4.11 | 24.39±4.15 | 24.02±3.90 | 24.18±4.49 | 23.36±3.99 | <0.001* |
| WC (cm) | 86.08±13.01 | 85.35±12.62 | 85.29±11.33 | 84.53±13.38 | 82.89±13.36 | <0.001* |
| DBP (mm Hg) | 77.07±12.07 | 77.28±12.53 | 77.01±12.51 | 76.31±12.80 | 74.72±25.82 | <0.001* |
| SBP (mm Hg) | 131.09±26.36 | 131.66±25.49 | 131.63±26.64 | 129.90±29.9 | 129.47±25.82 | 0.062* |

*P value for trend.

BMI, body mass index; CVD, cardiovascular disease; DBP, diastolic blood pressure; HbA$_1$c, glycosylated haemoglobin; HDL-C, high-density lipoprotein cholesterol; LDL-C, low-density lipoprotein cholesterol; SBP, systolic blood pressure; Scr, serum creatinine; SUA, serum uric acid; TG, triglyceride; WC, waist circle.

## RESULTS

The final sample consisted of 4513 women with an average age of 58.59±9.31 years (ranged from 45 years to 96 years). The mean and SD of menarche age were 16.27±2.18 years (ranged from 10 years to 30 years). The mean and SD of DBP and SBP level were 76.31±12.45 mm Hg (ranged from 39 mm Hg to 136 mm Hg) and 131.65±26.73 mm Hg (ranged from 76 mm Hg to 235 mm Hg), respectively. The mean and SD of BMI and WC level were 24.04±4.13 kg/m$^2$ (ranged from 11.65 kg/m$^2$ to 63.05 kg/m$^2$) and 84.67±12.83 cm (ranged from 19.5 cm to 130.10 cm), respectively. Table 1 showed the characteristics by the age of menarche. There were differences between the age of menarche in educational attainment, region, cigarette smoking, entertainment,and history of CVDs. However, between-group differences in the prevalence of marital status, alcohol consumption, eating habits, the experience of a traumatic event, physical activity, hepatitis history, antilipidaemic medication history and antidiabetic medication history were not found. Women with a higher age of menarche had higher HDL-C level and age but had lower TG, SUA, BMI, WC and DBP level. Moreover, there were no differences in LDL-C, LDL-C/HDL-C, HbA$_1$c, Scr and SBP level between the age of menarche groups.

Characteristics, according to BMI and WC in women, are presented in table 2. First, women in in rural places, more alcohol consumption and higher age of menarche had lower BMI and WC level. Second, women with a history of CVDs, antilipidaemic medication and antidiabetic medication had higher BMI and WC level than women without

medical illness. Third, women with higher educational attainment, being married, being never smoking, taking more eating meals and taking entertainment had higher BMI level. At last, women with the experience of a traumatic event had lower WC level.

Characteristics, according to DBP and SBP in women, are presented in table 3. First, women with more alcohol consumption had lower DBP and SBP levels. Second, women with a history of CVDs and antilipidaemic medication had higher DBP and SBP levels than women without medical illness. Third, women eating more meals, higher age of menarches had lower DBP level. At last, women with higher educational attainment, being married and having regular physical exercises had lower SBP level, but women with antidiabetic medication had higher SBP level than women without medical illness.

Correlations between characteristics, independent, moderators, mediators and dependent variables, according to DBP and SBP, are presented in table 4. First, drinking, history of CVDs, antilipidaemic medication, LDL-C, HDL-C, LDL-C/HDL-C, TG, HbA$_1$c, Scr, SUA, BMI and WC were significantly correlated with DBP and SBP in women. Second, eating habits, history of liver diseases and menarche age were significantly correlated with DBP. At last, age, educational attainment, marital status, physical exercises and antidiabetic medication were significantly correlated with SBP.

In the moderator analysis, we tested the interaction between menarche age and obesity in predicting systolic and diastolic BPs. The SE, β and the adjusting associated 95% CIs are shown in table 5. We controlled for

**Table 2** Characteristics of the participants with BMI and WC (N=4513)

| Variables | BMI (cm) | WC (cm) | P value for BMI | P value for WC |
|---|---|---|---|---|
| Educational attainment | | | | |
| Illiterate | 23.57±4.19 | 84.35±12.95 | 0.001* | 0.971* |
| Less than elementary school | 24.31±4.03 | 85.14±12.48 | | |
| High school | 24.93±4.07 | 83.17±14.65 | | |
| Above vocational school | 24.85±4.43 | 83.40±13.99 | | |
| Marital status | | | | |
| Single | 23.21±4.18 | 84.93±12.04 | <0.001 | 0.582 |
| Married | 24.19±4.10 | 84.63±12.97 | | |
| Region | | | | |
| Rural | 23.64±3.96 | 83.95±12.42 | <0.001 | <0.001 |
| Urban | 24.75±4.32 | 85.94±13.43 | | |
| Cigarette smoking | | | | |
| Never smoker | 24.11±4.12 | 84.73±12.77 | <0.001* | 0.135* |
| Previous smoker | 23.79±4.14 | 86.83±10.36 | | |
| Current smoker | 23.17±4.11 | 83.13±14.40 | | |
| Alcohol consumption | | | | |
| Never drinker | 24.10±4.14 | 85.02±12.5 | 0.011* | <0.001* |
| Less than once a month | 23.88±4.17 | 81.78±15.16 | | |
| More than once a month | 23.51±3.94 | 82.49±14.62 | | |
| Eating habits | | | | |
| ≤2 meals per day | 23.27±3.89 | 84.12±10.42 | <0.001* | 0.734* |
| 3 meals per day | 24.18±4.14 | 84.82±13.18 | | |
| ≥4 meals per day | 23.10±4.69 | 80.82±12.01 | | |
| Entertainment | | | | |
| No | 23.64±4.09 | 84.34±12.08 | <0.001 | 0.085 |
| Yes | 24.45±4.13 | 85.00±13.53 | | |
| Experience of a traumatic event | | | | |
| No | 24.06±4.14 | 84.82±12.57 | 0.474 | 0.004 |
| Yes | 23.88±3.94 | 82.64±15.81 | | |
| Physical activity | | | | |
| No physical exercise | 24.05±4.29 | 84.79±12.74 | 0.400* | 0.701* |
| Less than regular physical exercises | 23.82±3.70 | 84.25±12.48 | | |
| Regular physical exercises | 24.26±4.03 | 84.74±13.48 | | |
| History of CVDs | | | | |
| No | 23.92±4.09 | 84.26±12.63 | <0.001 | <0.001 |
| Yes | 24.83±4.28 | 87.25±13.89 | | |
| Hepatitis history | | | | |
| No | 24.04±4.15 | 84.66±12.79 | 0.871 | 0.948 |
| Yes | 24.09±3.47 | 84.73±14.37 | | |
| History of antilipidaemic medication | | | | |
| No | 23.87±3.99 | 84.22±12.72 | <0.001 | <0.001 |
| Yes | 26.75±5.14 | 91.59±12.58 | | |
| History of antidiabetic medication | | | | |
| No | 23.95±4.01 | 84.42±12.81 | <0.001 | <0.001 |
| Yes | 26.13±5.81 | 90.18±12.20 | | |

**Table 2** Continued

| Variables | BMI (cm) | WC (cm) | P value for BMI | P value for WC |
|---|---|---|---|---|
| Menarche age (years) | | | | |
| ≤14 | 24.63±4.11 | 86.08±13.01 | <0.001* | <0.001* |
| 15 | 24.39±4.15 | 85.35±12.62 | | |
| 16 | 24.02±3.90 | 85.29±11.33 | | |
| 17 | 24.18±4.49 | 84.53±13.38 | | |
| ≥18 | 23.36±13.36 | 82.89±13.36 | | |

*P value for trend.
BMI, body mass index; CVD, cardiovascular disease; WC, waist circle.

sociodemographic characteristics, health behaviours, medical histories and metabolic measures. After controlling for the total potential confounders, the moderating effect of obesity indicators on the relationship between age of menarche and BP was tested. The interaction term of obesity parameters×age of menarche was not significant for predicting either DBP (BMI, adjusted p=0.2556; WC, adjusted p=0.1833) or SBP (BMI, adjusted p=0.8561; WC, adjusted p=0.8427) in women.

In the mediation analysis, we estimated the BP equation using bootstrap inference for model coefficients. The B, β and the adjusting associated 95% CIs are shown in table 6. We controlled for sociodemographic characteristics, health behaviours, medical histories and metabolic measures. After controlling for the total potential confounders, the mediating effect of obesity indicators on the relationship between age of menarche and BP was tested. All correlations were significant correlation between age of menarche, obesity parameters and BP except the path of the age of menarche→SBP (BMI, crude p=0.9797; WC, crude p=0.9804) in model 1. In general, after controlling for potential confounders, BMI (DBP: adjusted p=0.0021, SBP: adjusted p<0.001) and WC (DBP: adjusted p=0.0020, SBP: adjusted p<0.001) partly mediated the relationship between the age of menarche and BP.

## DISCUSSION

In the study, we attempted to explore the association between age of menarche and BP and examined the moderating and mediating effect of obesity parameters (BMI and WC) on the association. We have found that the interaction term of obesity parameters×age of menarche was not significant for predicting BP. It was also important to note that WC and BMI partly mediated the association between age of menarche and BP. The results of our study were partly constant with most previous studies. So, previous studies had only focused on the relationship between earlier biological maturation and health outcomes in late adolescence and adulthood. Most of the previous studies meant exactly that: earlier biological maturation was at a higher risk of health outcomes in late adolescence and adulthood. Furthermore, most

previous studies found that early biological maturation (age of menarche) or puberty was significantly related to obesity parameters,[34–38] hypertension[38–41] and metabolic syndrome.[42–50] In general, our results were mostly consistent with the results of previous studies that suggest that earlier biological maturation is a risk factor for several negative health outcomes in late adolescence and adulthood.

Our study suggested the likelihood of a dose–response relationship between age of menarche and obesity parameters (BMI and WC). The relationship could be identified by the fact that early-maturing girls already have more adiposity before menarche, given that fatty tissue, through leptin release, has a crucial role in the initiation of the biological maturation process.[51 52] Early-maturing girls could maintain their higher fatty tissue in adulthood.[53] Another mechanism may also be associated with the time of the maturation process. It has been reported that early-maturing girls could experience a more accelerated maturation process than late-maturing ones. Thus, more significant interference with biological maturation in girls can lead to increased more fatty tissue.[54] We also found a significantly negative relationship between age at menarche and BP in middle-aged and elderly Chinese women. However, this association did not exist between the age of menarche and SBP before adjusting the related confounders. This result is consistent with previous findings[55–63] regarding the age of menarche as a cardiovascular risk factor. The weak association between age at menarche and BP in adulthood could, in part, be explained by the fact that early-maturers were younger than women with late menarche. This result could be an expression of the fact that BP is an outcome with a higher latency period than obesity. Moreover, among women who already entered menopause in our study, the effects of menarche could be even weaker, as results suggest, and obesity could have more effect on BP. We also found that women with a higher age of menarche had higher HDL-C levels but had lower DBP level. The results of the current study were quite consistent with those of Feng *et al*.[22] Increased body fat, especially abdominal fat, partially explained the increased insulin resistance and dyslipidaemia in women with early-menarche age. Therefore,

**Table 3** Characteristics of the participants with DBP and SBP (N=4513)

| Variables | DBP (mm Hg) | SBP (mm Hg) | P value for DBP | P value for SBP |
|---|---|---|---|---|
| Educational attainment | | | | |
| Illiterate | 76.37±12.45 | 133.93±28.43 | 0.288* | <0.001* |
| Less than elementary school | 76.38±12.42 | 128.89±24.99 | | |
| High school | 76.12±12.65 | 122.98±18.58 | | |
| Above vocational school | 73.86±12.47 | 127.84±40.38 | | |
| Marital status | | | | |
| Single | 76.70±11.96 | 139.61±31.93 | 0.388 | <0.001 |
| Married | 76.25±12.53 | 129.12±25.42 | | |
| Region | | | | |
| Rural | 76.20±12.72 | 130.18±25.92 | 0.435 | 0.121 |
| Urban | 76.50±11.96 | 131.46±28.06 | | |
| Cigarette smoking | | | | |
| Never smoker | 76.32±12.45 | 130.43±26.97 | 0.673* | 0.051* |
| Previous smoker | 73.60±11.41 | 132.13±22.83 | | |
| Current smoker | 77.09±12.65 | 133.61±23.78 | | |
| Alcohol consumption | | | | |
| Never drinker | 76.53±12.38 | 131.27±27.27 | 0.003* | 0.001* |
| Less than once a month | 74.76±12.18 | 123.81±19.94 | | |
| More than once a month | 74.73±13.26 | 127.77±23.06 | | |
| Eating habits | | | | |
| ≤2 meals per day | 77.69±12.60 | 132.14±26.68 | <0.001* | 0.117* |
| 3 meals per day | 76.17±12.41 | 130.44±26.37 | | |
| ≥4 meals per day | 71.56±11.46 | 128.63±44.41 | | |
| Entertainment | | | | |
| No | 76.08±12.60 | 130.66±26.33 | 0.216 | 0.975 |
| Yes | 76.54±12.30 | 130.64±27.12 | | |
| Experience of a traumatic event | | | | |
| No | 76.38±12.47 | 130.84±27.12 | 0.219 | 0.075 |
| Yes | 75.48±12.14 | 128.05±20.66 | | |
| Physical activity | | | | |
| No physical exercise | 76.34±12.46 | 131.47±28.25 | 0.873* | 0.024* |
| Less than regular physical exercises | 76.05±12.76 | 129.23±22.14 | | |
| Regular physical exercises | 76.50±12.08 | 129.54±26.03 | | |
| History of CVDs | | | | |
| No | 76.11±12.44 | 130.02±26.29 | 0.021 | <0.001 |
| Yes | 77.39±12.40 | 134.50±29.28 | | |
| Hepatitis history | | | | |
| No | 76.36±12.45 | 130.76±26.50 | 0.017 | 0.144 |
| Yes | 74.10±11.56 | 127.76±32.79 | | |
| History of antilipidaemic medication | | | | |
| No | 76.07±12.34 | 130.12±26.01 | <0.001 | <0.001 |
| Yes | 80.09±13.46 | 138.80±35.01 | | |
| History of antidiabetic medication | | | | |
| No | 76.25±12.47 | 130.17±25.46 | 0.100 | <0.001 |
| Yes | 77.73±11.83 | 140.96±45.17 | | |

**Table 3** Continued

| Variables | DBP (mm Hg) | SBP (mm Hg) | P value for DBP | P value for SBP |
|---|---|---|---|---|
| Menarche age (years) | | | | |
| ≤14 | 77.07±12.07 | 131.09±26.36 | | |
| 15 | 77.28±12.53 | 131.66±25.49 | <0.001* | 0.062* |
| 16 | 77.01±12.51 | 131.63±26.64 | | |
| 17 | 76.31±12.80 | 129.90±29.90 | | |
| ≥18 | 74.72±12.31 | 129.47±25.82 | | |

*P value for trend.
CVD, cardiovascular disease; DBP, diastolic blood pressure; SBP, systolic blood pressure.

**Table 4** Relationship between various characteristics and blood pressure status of the participants (N=4513)

| Variables | DBP R (P value) | SBP R (P value) |
|---|---|---|
| Age | −0.016 (0.275) | 0.259 (<0.001) |
| Educational attainment | −0.016 (0.288) | −0.106 (<0.001) |
| Marital status | −0.013 (0.388) | −0.139 (<0.001) |
| Current residence | 0.012 (0.435) | 0.023 (0.121) |
| Smoking status | 0.006 (0.673) | 0.029 (0.051) |
| Drinking | −0.045 (0.003) | −0.051 (0.001) |
| Eating habits | −0.055 (<0.001) | −0.023 (0.117) |
| Entertainment | 0.018 (0.216) | 0.000 (0.975) |
| Experience of a traumatic event | −0.018 (0.219) | −0.027 (0.075) |
| Physical exercises | 0.002 (0.873) | −0.034 (0.024) |
| History of CVDs | 0.035 (0.018) | 0.058 (<0.001) |
| History of liver diseases | −0.035 (0.018) | −0.022 (0.146) |
| Antilipidaemic medication | 0.078 (<0.001) | 0.078 (<0.001) |
| Antidiabetic medication | 0.025 (0.100) | 0.083 (<0.001) |
| LDL-C | 0.050 (0.001) | 0.060 (<0.001) |
| HDL-C | −0.099 (<0.001) | −0.062 (<0.001) |
| LDL-C/HDL-C | 0.106 (<0.001) | 0.094 (<0.001) |
| TG | 0.100 (<0.001) | 0.091 (<0.001) |
| HbA$_1$c | 0.061 (<0.001) | 0.066 (<0.001) |
| Scr | 0.056 (<0.001) | 0.076 (<0.001) |
| SUA | 0.085 (<0.001) | 0.111 (<0.001) |
| BMI | 0.221 (<0.001) | 0.129 (<0.001) |
| WC | 0.183 (<0.001) | 0.177 (<0.001) |
| Menarche age | −0.060 (<0.001) | −0.014 (0.335) |

BMI, body mass index; CVD, cardiovascular disease; DBP, diastolic blood pressure; HbA$_1$c, glycosylated haemoglobin; HDL-C, high-density lipoprotein cholesterol; LDL-C, low-density lipoprotein cholesterol; SBP, systolic blood pressure; Scr, serum creatinine; SUA, serum uric acid; TG, triglyceride; WC, waist circle.

obesity in adolescence, sociodemographic or even behavioural factors through life (including age, educational attainment, marital status, region, smoking status, alcohol consumption, eating habits, entertainment, the experience of a traumatic event and physical activity) were potential mediators that were most important for BP control and management.

Though so many studies have explored the association analysis between the age of menarche or obesity parameters and BP, there were only two studies[64 65] that explored the mediating effect of obesity parameters on the relation between age of menarche and BP. Zhang et al[64] found that the relationship between age of menarche and DBP was partly mediated by WC. In contrast, the relationship between age of menarche and SBP was fully mediated by WC in women. Werneck et al[65] showed that women with late menarche are less likely to have a risk of hypertension, and BMI was an important mediator of the age in the menarche–hypertension association. Interestingly, we found that obesity parameters partly mediated the relationship between the age of menarche and BP in women. The difference between our research and others may due to the different populations, different definitions of early and later menarche, and different confounding variables by controlling. The individuals in our study were middle-aged and elderly Chinese women, where the mean age at recruitment and age of menarche were older than those in Werneck et al's study,[65] and the level of socio-economic development also made some contribution to ontogenetic development. In addition, other studies and a meta-analysis[55 57 63 66–68] reported an inverse association between early age of menarche and hypertension, which was not observed in Zhang et al's study.[64] They found no direct relationship between the age of menarche and DBP, and we have found a direct relationship between the age of menarche and DBP. The participants were similar in China and its socioeconomic background; this phenomenon could be explained by the cumulative effect, which showed a significantly negative association between age of menarche and DBP over life span.

**Table 5** Moderator effect of obesity on the relationship between menarche age and blood pressure in women (N=4513)

| Dependent | Variables | Model 1* | | | Model 2† | | | Model 3‡ | | | Model 4§ | | | Model 5‖ | | |
|---|---|---|---|---|---|---|---|---|---|---|---|---|---|---|---|---|
| | | SE | B (95% CI) | P value | SE | B (95% CI) | P value | SE | B (95% CI) | P value | SE | B (95% CI) | P value | SE | B (95% CI) | P value |
| DBP | Menarche age | 0.0835 | −0.2027 (−0.3665 to −0.0390) | 0.0153 | 0.0855 | −0.2528 (−0.4203 to −0.0852) | 0.0031 | 0.0853 | −0.2411 (−0.4084 to −0.0738) | 0.0047 | 0.0854 | −0.2485 (−0.4158 to −0.0811) | 0.0036 | 0.0973 | −0.2162 (−0.4069 to −0.0254) | 0.0263 |
| | BMI | 0.0440 | 0.6546 (0.5684 to 0.7409) | <0.001 | 0.0447 | 0.6757 (0.5881 to 0.7634) | <0.001 | 0.0449 | 0.6835 (0.5955 to 0.7715) | <0.001 | 0.0458 | 0.6647 (0.5750 to 0.7544) | <0.001 | 0.0556 | 0.5916 (0.4827 to 0.7005) | <0.001 |
| | BMI×menarche age | 0.0204 | 0.0279 (−0.0121 to 0.0678) | 0.1719 | 0.0204 | 0.0258 (−0.0141 to 0.0658) | 0.2046 | 0.0203 | 0.0247 (−0.0152 to 0.0645) | 0.2248 | 0.0203 | 0.0220 (−0.0178 to 0.0618) | 0.2790 | 0.0229 | 0.0260 (−0.0189 to 0.071) | 0.2556 |
| SBP | Menarche age | 0.1826 | −0.0003 (−0.3583 to 0.3576) | 0.9986 | 0.1789 | −0.5822 (−0.9329 to −0.2315) | 0.0011 | 0.1788 | −0.5682 (−0.9187 to −0.2177) | 0.0015 | 0.1795 | −0.5944 (−0.9462 to −0.2425) | <0.001 | 0.2141 | −0.5084 (−0.9282 to −0.0886) | 0.0176 |
| | BMI | 0.0964 | 0.8361 (0.6470 to 1.0251) | <0.001 | 0.0938 | 1.0927 (0.9088 to 1.2766) | <0.001 | 0.0942 | 1.0906 (0.9058 to 1.2753) | <0.001 | 0.0965 | 1.0287 (0.8396 to 1.2178) | <0.001 | 0.1223 | 0.8846 (0.6448 to 1.1244) | <0.001 |
| | BMI×menarche age | 0.0446 | 0.0178 (−0.0696 to 0.1052) | 0.6896 | 0.0426 | 0.0111 (−0.0726 to 0.0947) | 0.7955 | 0.0426 | 0.0113 (−0.0722 to 0.0948) | 0.7914 | 0.0427 | 0.0052 (−0.0786 to 0.0889) | 0.9038 | 0.0504 | 0.0091 (−0.0897 to 0.108) | 0.8561 |
| DBP | Menarche age | 0.0838 | −0.2569 (−0.4213 to −0.0925) | 0.0022 | 0.0861 | −0.2588 (−0.4277 to −0.0900) | 0.0027 | 0.0861 | −0.2481 (−0.4168 to −0.0794) | 0.0040 | 0.0861 | −0.2561 (−0.4248 to −0.0873) | 0.0029 | 0.0979 | −0.2001 (−0.3921 to −0.0082) | 0.0410 |
| | WC | 0.0142 | 0.1732 (0.1452 to 0.2011) | <0.001 | 0.0143 | 0.1752 (0.1471 to 0.2032) | <0.001 | 0.0143 | 0.1730 (0.1449 to 0.2011) | <0.001 | 0.0145 | 0.1645 (0.1361 to 0.193) | <0.001 | 0.0172 | 0.1472 (0.1135 to 0.1809) | <0.001 |
| | WC×menarche age | 0.0065 | 0.0058 (−0.007 to 0.0186) | 0.3758 | 0.0066 | 0.0054 (−0.0075 to 0.0182) | 0.4136 | 0.0065 | 0.005 (−0.0079 to 0.0178) | 0.4473 | 0.0065 | 0.0043 (−0.0085 to 0.0171) | 0.5075 | 0.0074 | 0.0099 (−0.0047 to 0.0244) | 0.1833 |
| SBP | Menarche age | 0.1777 | −0.0009 (−0.3492 to 0.3475) | 0.9961 | 0.1761 | −0.5656 (−0.9107 to −0.2204) | 0.0013 | 0.1760 | −0.5546 (−0.8997 to −0.2094) | 0.0016 | 0.1766 | −0.581 (−0.9273 to −0.2347) | 0.0010 | 0.2096 | −0.4548 (−0.8658 to −0.0438) | 0.0301 |
| | WC | 0.0302 | 0.3637 (0.3044 to 0.4229) | <0.001 | 0.0293 | 0.3228 (0.2654 to 0.3802) | <0.001 | 0.0294 | 0.3153 (0.2578 to 0.3729) | <0.001 | 0.0298 | 0.2944 (0.236 to 0.3528) | <0.001 | 0.0368 | 0.2744 (0.2022 to 0.3466) | <0.001 |
| | WC×menarche age | 0.0139 | −0.0156 (−0.0428 to 0.0116) | 0.2607 | 0.0134 | −0.0061 (−0.0323 to 0.0202) | 0.6501 | 0.0134 | −0.0063 (−0.0326 to 0.0199) | 0.6367 | 0.0134 | −0.0081 (−0.0344 to 0.0182) | 0.5443 | 0.0159 | −0.0032 (−0.0343 to 0.028) | 0.8427 |

*Unadjusted.
†Adjusted for age, educational attainment, marital status and region.
‡Adjusted for age, educational attainment, marital status, region, smoking status, alcohol consumption, eating habits, entertainment, experience of a traumatic event and physical activity.
§Adjusted for age, educational attainment, marital status, region, smoking status, alcohol consumption, eating habits, entertainment, experience of a traumatic event, physical activity, history of CVDs, hepatitis history, antihyperlipidaemic drugs and antidiabetic drugs.
‖Adjusted for age, educational attainment, marital status, region, smoking status, alcohol consumption, eating habits, entertainment, experience of a traumatic event, physical activity, history of CVDs, hepatitis history, antihyperlipidaemic drugs, antidiabetic drugs, LDL–C, HDL–C, TG, HbA1c, Scr and SUA.
BMI, body mass index; CVD, cardiovascular disease; DBP, diastolic blood pressure; HbA1c, haemoglobin A1C; HDL–C, high-density lipoprotein cholesterol; LDL–C, low-density lipoprotein cholesterol; SBP, systolic blood pressure; Scr, serum creatinine; SUA, serum uric acid; TG, triglycerides; WC, waist circle.

**Table 6** Mediation effect of obesity on the relationship between age at menarche and blood pressure in women (N=4513)

| Path | | Model 1* | | | Model 2† | | | Model 3‡ | | | Model 4§ | | | Model 5¶ | | |
|---|---|---|---|---|---|---|---|---|---|---|---|---|---|---|---|---|
| | | β | B (95% CI) | P value | β | B (95% CI) | P value | β | B (95% CI) | P value | β | B (95% CI) | P value | β | B (95% CI) | P value |
| 1 | Menarche age→BMI | −0.1080 | −0.2045 (−0.2595 to −0.1496) | <0.001 | −0.0725 | −0.1374 (−0.1930 to −0.0818) | <0.001 | −0.0707 | −0.1339 (−0.1892 to −0.0786) | <0.001 | −0.0705 | −0.1334 (−0.1880 to −0.0788) | <0.001 | −0.0468 | −0.0885 (−0.1409 to −0.0361) | <0.001 |
| | Menarche age→DBP | −0.0367 | −0.2096 (−0.3731 to −0.0461) | 0.0120 | −0.0454 | −0.2596 (−0.4268 to −0.0924) | 0.0024 | −0.0433 | −0.2475 (−0.4145 to −0.0806) | 0.0037 | −0.0447 | −0.2542 (−0.4213 to −0.0872) | 0.0029 | −0.0379 | −0.2154 (−0.3833 to −0.0474) | 0.0120 |
| | BMI→DBP | 0.2171 | 0.6545 (0.5682 to 0.7408) | <0.001 | 0.2241 | 0.6757 (0.5881 to 0.7634) | <0.001 | 0.2276 | 0.6835 (0.5955 to 0.7715) | <0.001 | 0.2208 | 0.6643 (0.5746 to 0.7541) | <0.001 | 0.1986 | 0.5973 (0.5026 to 0.6921) | <0.001 |
| | Total effect | −0.0601 | −0.3435 (−0.5099 to −0.177) | <0.001 | −0.0617 | −0.3524 (−0.5234 to −0.1815) | <0.001 | −0.0593 | −0.3390 (−0.5098 to −0.1683) | <0.001 | −0.0602 | −0.3429 (−0.5133 to −0.1724) | <0.001 | −0.0471 | −0.2682 (−0.4388 to −0.0976) | 0.0021 |
| 2 | Menarche age→BMI | 0.1085 | −0.2051 (−0.2601 to −0.1502) | <0.001 | −0.0731 | −0.1381 (−0.1936 to −0.0826) | <0.001 | −0.0712 | −0.1346 (−0.1898 to −0.0793) | <0.001 | −0.0711 | −0.1342 (−0.1887 to −0.0797) | <0.001 | −0.0473 | −0.0892 (−0.1416 to −0.0369) | <0.001 |
| | Menarche age→SBP | −0.0004 | −0.0046 (−0.3619 to 0.3526) | 0.9797 | −0.0477 | −0.585 (−0.9350 to −0.2351) | 0.0011 | −0.0466 | −0.5711 (−0.9209 to −0.2213) | 0.0014 | −0.0486 | −0.5957 (−0.9468 to −0.2446) | <0.001 | −0.0446 | −0.5479 (−0.9025 to −0.1932) | 0.0025 |
| | BMI→SBP | 0.1289 | 0.8359 (0.6469 to 1.0249) | <0.001 | 0.1685 | 1.0927 (0.9088 to 1.2766) | <0.001 | 0.1682 | 1.0905 (0.9058 to 1.2753) | <0.001 | 0.1585 | 1.0286 (0.8395 to 1.2176) | <0.001 | 0.1441 | 0.9366 (0.7361 to 1.1371) | <0.001 |
| | Total effect | −0.0144 | −0.1761 (−0.5342 to 0.182) | 0.3350 | −0.0601 | −0.736 (−1.0902 to −0.3817) | <0.001 | −0.0586 | −0.7178 (−1.0719 to −0.3638) | <0.001 | −0.0599 | −0.7337 (−1.0883 to −0.3791) | <0.001 | −0.0515 | −0.6314 (−0.9889 to −0.2739) | <0.001 |
| 3 | Menarche age→WC | −0.0837 | −0.4928 (−0.6642 to −0.3213) | <0.001 | −0.0910 | −0.5355 (−0.7107 to −0.3602) | <0.001 | −0.0896 | −0.5275 (−0.7024 to −0.3526) | <0.001 | −0.0898 | −0.5285 (−0.7026 to −0.3545) | <0.001 | −0.0696 | −0.4085 (−0.5778 to −0.2392) | <0.001 |
| | Menarche age→DBP | −0.0454 | −0.2589 (−0.4232 to −0.0945) | 0.0020 | −0.0456 | −0.26 (−0.4288 to −0.0912) | 0.0025 | −0.0437 | −0.2491 (−0.4178 to −0.0805) | 0.0038 | −0.0452 | −0.2571 (−0.4257 to −0.0884) | 0.0028 | −0.0373 | −0.2118 (−0.3811 to −0.0424) | 0.0143 |
| | WC→DBP | 0.1789 | 0.1733 (0.1454 to 0.2013) | <0.001 | −0.1810 | 0.1754 (0.1473 to 0.2034) | <0.001 | 0.1787 | 0.1732 (0.1451 to 0.2013) | <0.001 | 0.1705 | 0.1647 (0.1363 to 0.1931) | <0.001 | 0.1445 | 0.1399 (0.1104 to 0.1695) | <0.001 |
| | Total effect | −0.0604 | −0.3443 (−0.5107 to −0.1779) | <0.001 | −0.0620 | −0.3539 (−0.5248 to −0.183) | <0.001 | −0.0597 | −0.3405 (−0.5112 to −0.1698) | <0.001 | −0.0606 | −0.3441 (−0.5145 to −0.1737) | <0.001 | −0.0474 | −0.2689 (−0.4395 to −0.0984) | 0.0020 |
| 4 | Menarche age→WC | −0.0849 | −0.4944 (−0.6658 to −0.323) | <0.001 | −0.0914 | −0.5376 (−0.7129 to −0.3624) | <0.001 | −0.0901 | −0.5299 (−0.7048 to −0.355) | <0.001 | −0.0904 | −0.531 (−0.7051 to −0.357) | <0.001 | −0.0702 | −0.411 (−0.5803 to −0.2417) | <0.001 |
| | Menarche age→SBP | 0.0004 | 0.0044 (−0.3439 to 0.3526) | 0.9804 | −0.0468 | −0.5643 (−0.9094 to −0.2192) | 0.0014 | −0.0459 | −0.5533 (−0.8983 to −0.2082) | 0.0017 | −0.0481 | −0.5792 (−0.9255 to −0.233) | 0.0010 | −0.0433 | −0.5228 (−0.8721 to −0.1735) | 0.0034 |
| | WC→SBP | 0.1772 | 0.3632 (0.3040 to 0.4224) | <0.001 | 0.1574 | 0.3226 (0.2652 to 0.38) | <0.001 | 0.1538 | 0.3151 (0.2576 to 0.3727) | <0.001 | 0.1436 | 0.2942 (0.2358 to 0.3526) | <0.001 | 0.1290 | 0.2655 (0.2045 to 0.3266) | <0.001 |

Continued

**Table 6** Continued

| Path | Model 1* | | | Model 2† | | | Model 3‡ | | | Model 4§ | | | Model 5¶ | | |
|---|---|---|---|---|---|---|---|---|---|---|---|---|---|---|---|
| | β | B (95% CI) | P value | β | B (95% CI) | P value | β | B (95% CI) | P value | β | B (95% CI) | P value | β | B (95% CI) | P value |
| Total effect | -0.0145 | -0.1752 (-0.5277 to 0.1773) | 0.3300 | -0.0612 | -0.7377 (-1.086 to -0.3894) | <0.001 | -0.0598 | -0.7202 (-1.0683 to -0.3722) | <0.001 | -0.0611 | -0.7354 (-1.084 to -0.3869) | <0.001 | -0.0524 | -0.6320 (-0.9832 to -0.2807) | <0.001 |

Path 1: mediation effect of BMI on the relationship between age at menarche and DBP.
Path 2: mediation effect of BMI on the relationship between age at menarche and SBP.
Path 3: mediation effect of WC on the relationship between age at menarche and DBP.
Path 4: mediation effect of WC on the relationship between age at menarche and SBP.
*Unadjusted.
†Adjusted for age, educational attainment, marital status and region.
‡Adjusted for age, educational attainment, marital status, region, cigarette smoking, alcohol consumption, eating habits, entertainment, the experience of a traumatic event and physical activity.
§Adjusted for age, educational attainment, marital status, region, cigarette smoking, alcohol consumption, eating habits, entertainment, the experience of a traumatic event, physical activity, history of CVDs, hepatitis history, history of antilipidaemic medication and history of antidiabetic medication.
¶Adjusted for age, educational attainment, marital status, region, cigarette smoking, alcohol consumption, eating habits, entertainment, the experience of a traumatic event, physical activity, history of CVDs, hepatitis history, history of antilipidaemic medication, history of antidiabetic medication, LDL-C, HDL-C, TG, HbA1c, Scr and SUA.
B, unstandardised coefficient; BMI, body mass index; CVD, cardiovascular disease; DBP, diastolic blood pressure; HbA1c, haemoglobin A1C; HDL-C, high-density lipoprotein cholesterol; LDL-C, low-density lipoprotein cholesterol; SBP, systolic blood pressure; Scr, serum creatinine; SUA, serum uric acid; TG, triglyceride; WC, waist circle; β, standardised coefficient.

There were several limitations to the study. The main limitations of our research are related to the cross-sectional study and the self-reported method used for the assessment of age at menarche and most related confounders. BP in our study was measured at home by professionally trained volunteers, and the next step is the use of clinical BP and ambulatory BP. However, this is the first large population study to examine the moderation between the age of menarche and obesity parameters in predicting BP in middle-aged and elderly Chinese, as well as the mediation effects of obesity parameters on the relationship between age of menarche and BP. Moreover, a significant strength of the study is a large sample of 4513 middle-aged and older Chinese. The results presented in this article represent the baseline data that could be explored further in prospective cohort studies.

In sum, the interaction term of obesity parameters×age of menarche was not significant for predicting either DBP or SBP in women. Moreover, obesity parameters partly mediated the relationship between the age of menarche and BP.

## CONCLUSIONS

The interaction term of obesity parameters×age of menarche was not significant for predicting either DBP or SBP in women. Moreover, obesity parameters partly mediated the relationship between age of menarche and BP.

**Author affiliations**
[1]Department of Internal Medicine Nursing, School of Nursing, Wannan Medical College, Wuhu, Anhui, China
[2]Obstetrics and Gynecology Nursing, School of Nursing, Wannan Medical College, Wuhu, Anhui, China
[3]Department of Pediatric Nursing, School of Nursing, Wannan Medical College, Wuhu, Anhui, China
[4]Department of Surgical Nursing, School of Nursing, Wannan Medical College, Wuhu, Anhui, China
[5]School of Nursing, Henan University of Science and Technology, Luoyang, Henan, China
[6]Department of Emergency and Critical Care Nursing, School of Nursing, Wannan Medical College, Wuhu, Anhui, China
[7]Rehabilitation Nursing, School of Nursing, Wannan Medical College, Wuhu, Anhui, China
[8]Department of Nursing, Yijishan Hospital, the First Affiliated Hospital of Wannan Medical College, Wuhu, Anhui, China

**Acknowledgements** We are grateful to the China Center for Economic Research, and we also thank all participants for their contribution and members of the research for collecting the data.

**Contributors** Conceived and designed the research, wrote the paper and analysed the data: LZ. Revised the paper: LZ, LY, CW, TY, DZ, HW, JL, YLe, LS, XL, YH, HC and YLi. LZ as the guarantor was responsible for the overall content.

**Funding** China Health and Retirement Longitudinal Study was supported by the NSFC (70910107022 and 71130002) and National Institute on Aging (R03-TW008358-01 and R01-AG037031-03S1) and World Bank (7159234), and the publication fee was supported by the Support Program for Outstanding Young Talents from the Universities and Colleges of Anhui Province for Lin Zhang (gxyqZD2021118).

**Competing interests** None declared.

**Patient and public involvement** Patients and/or the public were not involved in the design, conduct, reporting or dissemination plans of this research.

**Patient consent for publication** Not applicable.

**Ethics approval** This study involves human participants and was approved by the ethics review committee of Peking University. All participants provided informed consent. Approval for this study was given by the medical ethics committee of Wannan Medical College (approval number 2021–3). The participants gave informed consent to participate in the study before taking part.

**Provenance and peer review** Not commissioned; externally peer reviewed.

**Data availability statement** Data are available in a public, open access repository. Data can be accessed via http://charls.pku.edu.cn/.

**Open access** This is an open access article distributed in accordance with the Creative Commons Attribution 4.0 Unported (CC BY 4.0) license, which permits others to copy, redistribute, remix, transform and build upon this work for any purpose, provided the original work is properly cited, a link to the licence is given, and indication of whether changes were made. See: https://creativecommons.org/licenses/by/4.0/.

**ORCID iD**
Lin Zhang http://orcid.org/0000-0003-4318-5060

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
