## [Reviewer comments · BMJ Open]

ARTICLE DETAILS

TITLE (PROVISIONAL)	Mediator or moderator? The role of obesity in the association between age at menarche and blood pressure in mid-aged and elderly Chinese: a population-based cross-sectional study
AUTHORS	zhang, lin; Yang, Liu; Wang, Congzhi; Yuan, Ting; Zhang, Dongmei; Wei, Huanhuan; Li, Jing; Lei, Yunxiao; Sun, Lu; Li, Xiaoping; Hua, Ying; Che, Hengying; Li, Yuanzhen

VERSION 1 – REVIEW

REVIEWER	Al-Ahmad, Basma International Islamic University Malaysia, fundamental dental and medical basic medical science
REVIEW RETURNED	17-Jun-2021

GENERAL COMMENTS	1. To enhance the importance of the manuscript, the authors are advised to justify the merit of the study.2. To improve the validity of the statistical analysis, the authors are requested to show how the sample size was determined.3. The research methods were appropriately conducted. However, some parts of the methodology are not fully described and need further explanation.(like the lipid profile measurement , did you measure the ratio of LDL to HDL ? if you did please add it to the results)4. page 9 line 25 , and page 11 line 25 you mentioned HCL-C ? what do you mean is it HDL-C ? if not please explain what do you mean .5. page 19,20 Discussion : please add further discussion and explanation about the relation between woman with higher age of menarche with the changes in the lipid and blood pressure that you already highlight in the results page 11 line 25,26.
--

REVIEWER	Jaruratanasirikul, Somchit Prince Songkla Univ, Pediatrics
REVIEW RETURNED	27-Jul-2021

GENERAL COMMENTS	This cross-sectional study included 4513 Chinese women, aged 45-96, recruited from the Chinese Health and Retirement longitudinal Study. The objective of this study was to determine the correlation/interaction of the age at menarche, BMI and hypertension in elderly age group, using moderator analysis and mediator analysis. The results showed that obesity parameters and age at menarche were not significant to predict hypertension, but obesity was partly related to the age at menarche and diastolic hypertension. Comment 1. What is the exact number of initially recruited women at Wave 1? Was it 6883 (page 6) or 8779 (page 7)?
--

	2. Ethical approval and Informed consent were not mentioned in the Method Section. 3. What was the measurement method of blood pressure, various measurement by various personnel? This can be discussed as a limitation of this study. 4. What was the method of height measurement, particularly in elderly women who might have kyphoscoliosis? Was the height measured or just self-reported? 5. There were wrong spelling words such as HCL-C (page 9), studier (page 17). 6. The descriptive legends of Tables 1, 2 and 3 were the same!!! The authors should be careful for these details
--	---

VERSION 1 – AUTHOR RESPONSE

Reviewer: 1

Dr. Basma Al-Ahmad, International Islamic University Malaysia

Comments to the Author:

1. To enhance the importance of the manuscript, the authors are advised to justify the merit of the study.

Response: thank you very much, we have revised the discussion, and added the strength to the study.

There were several limitations to the study. The main limitations of our research are related to the cross-sectional study and the self-reported method used for the assessment of age at menarche and most related confounders. Blood pressure in our study was measured at home by professionally trained volunteers, and the next step is the use of clinical blood pressure and ambulatory blood pressure. **However, this is the first large population study to examine the moderation between the age of menarche and obesity parameters in predicting blood pressure (BP) in middle-aged and elderly Chinese, as well as the mediation effects of obesity parameters on the relationship between age of menarche and BP. Moreover, a significant strength of the study is a large sample of 4513 middle-aged and older Chinese. The results provided baseline data that could be explored further in future prospective studies.**

2. To improve the validity of the statistical analysis, the authors are requested to show how the sample size was determined.

Response: thank you very much, we have revised the sample size.

Individuals

The Individuals of the study were selected from the China Health and Retirement Longitudinal Study (CHARLS), Wave 1 (2011). At baseline, 6883 women were recruited for a longitudinal study, 1875 individuals were excluded because the absence of metabolic measures, a group of 41 participants did not have medication history, and 65 individuals did not have their blood pressure, 389 individuals used antihypertension drugs. Finally, 4513 individuals were included in the analyses. Figure 1 summarized the selection of participants. Our study involved 4513 individuals aged ≥ 45 years were total women [mean \pm standard deviation age= 58.59 ± 9.31 years, ranged from 45 to 96years]. The mean and standard deviation of menarche age were 16.27 ± 2.18 years (ranged from 10 to 30 years).

Fig. 1 Selection of participants

3. The research methods were appropriately conducted. However, some parts of the methodology are not fully described and need further explanation.(like the lipid profile measurement , did you measure the ratio of LDL to HDL ? if you did please add it to the results)

Response: thank you very much, we have added the **LDL-C/HDL-C** to the result, thanks again.

4. page 9 line 25 , and page 11 line 25 you mentioned HCL-C ? what do you mean is it HDL-C ? if not please explain what do you mean .

Response: thank you very much, it was typo. It is HDL-C, and we have corrected the error. Thanks again.

5. page 19,20 Discussion : please add further discussion and explanation about the relation between woman with higher age of menarche with the changes in the lipid and blood pressure that you already highlight in the results page 11 line 25,26.

Response: thank you very much, we have revised the discussion, and added further discussion and explanation about the relation.

We also found a significantly negative relationship between age at menarche and BP in mid-aged and elderly Chinese women. However, this association did not exist between the age of menarche and SBP before adjusting the related confounders. This result consistent with previous findings regarding the age of menarche as a cardiovascular risk factor. The weak association between age at menarche and BP in adulthood could, in part, be explained by the fact that early matures were younger than women with late menarche. This result could be an expression of the fact that BP is an outcome with a higher latency period than obesity. Moreover, among women who already entered menopause in our study, the effects of menarche could be even weaker, as results suggest, and obesity could have more effect on BP. We also found that women with a higher age of menarche had higher HDL-C level but had lower DBP levels. The results of the current study were quite consistent with those of Feng Y, Hong X, Wilker E, et al. Increased body fat, especially abdominal fat, partially explained the increased insulin resistance and dyslipidemia in women with early menarche age. Therefore, obesity in childhood/adolescence, sociodemographic, or even behavioral factors through life (including age, educational levels, marital status, place of residence, smoking habits, drinking habits, eating meals, social and leisure activities, the experience of a traumatic event, taking physical activity or exercise) were potential mediators to target when interventions for BP control and management.

Dr. Somchit Jaruratanasirikul, Prince Songkla Univ

Comments to the Author:

This cross-sectional study included 4513 Chinese women, aged 45-96, recruited from the Chinese Health and Retirement longitudinal Study. The objective of this study was to determine the correlation/interaction of the age at menarche, BMI and hypertension in elderly age group, using moderator analysis and mediator analysis. The results showed that obesity parameters and age at menarche were not significant to predict hypertension, but obesity was partly related to the age at menarche and diastolic hypertension.

Comment

1. What is the exact number of initially recruited women at Wave 1? Was it 6883 (page 6) or 8779 (page 7)?

Response: thank you very much, we have revised the sample size, and put detailed description on sample selection and data source.

The Individuals of the study were selected from the China Health and Retirement Longitudinal Study (CHARLS), Wave 1 (2011). At baseline, 6883 women were recruited for a longitudinal study, 1875 individuals were excluded because the absence of metabolic measures, a group of 41 participants did not have medication history, and 65 individuals did not have their blood pressure, 389 individuals used antihypertension drugs. Finally, 4513 individuals were included in the analyses. Figure 1 summarized the selection of participants. Our study involved 4513 individuals aged ≥ 45 years were total women [mean \pm standard deviation age = 58.59 ± 9.31 years, ranged from 45 to 96 years]. The mean and standard deviation of menarche age were 16.27 ± 2.18 years (ranged from 10 to 30 years).

Fig. 1 Selection of participants

2. Ethical approval and Informed consent were not mentioned in the Method Section.

Response: thank you very much, we have revised the related content in the method section, thanks again.

Study design and setting

Data from a longitudinal study named the China Health and Retirement Longitudinal Study (CHARLS) were used in our research. The CHARLS was a nationally representative longitudinal study conducted by the China Centre for Economic Research at Peking University from 2011. **The study was approved by the institutional ethical committees of Peking University. All participants provided informed consent.**

3. What was the measurement method of blood pressure, various measurement by various personnel? This can be discussed as a limitation of this study.

Response: thank you very much. Blood pressure in our study was measured at home by professionally trained volunteers. The average value of blood pressure was measured by the mean of the 3-time measurements. Performance measurements available in order to help decision makers choose the right ones for their specific purposes. There are usually three ways to measure blood pressure, Family self-test blood pressure, clinical blood pressure, and ambulatory blood pressure. And ambulatory blood pressure is the most accurate. However, the blood pressure measurement in our study is suitable for large-scale epidemiological investigation, and we add the limitation in our study.

4. What was the method of height measurement, particularly in elderly women who might have? Was the height measured or just self-reported?

Response: thank you very much, we have detailed the method of height measurement. The method was used in our previous study [23]. Thanks again.

Height was measured using a height measurement instrument (Omron hem-7200 sphygmomanometer). Participants were asked to stand barefoot on the base plate of the height meter in the upright position, with the back against the vertical back plate of the measuring instrument, with eyes open, arms alongside the body and heels were closed and the toes were separated by 60 degrees. Anyone who appeared to be unable to complete measurement in time due to health reasons such as hunchback and inability to stand was excluded from our study. The method was used in our previous study [23].

5. There were wrong spelling words such as HCL-C (page 9), studier (page 17).

Response: thank you very much, it was typo. It is HDL-C, and we have corrected the error. Thanks again.

6. The descriptive legends of Tables 1, 2 and 3 were the same!!! The authors should be careful for these details

Response: thank you very much, we have revised the legends of Tables, thanks again.

VERSION 2 – REVIEW

REVIEWER	Al-Ahmad, Basma International Islamic University Malaysia, fundamental dental and medical basic medical science
REVIEW RETURNED	20-Dec-2021

GENERAL COMMENTS	The author already follow all the instructions and did all the requested changes that have been send to him previously, changes have been done accordingly
--

REVIEWER	Jaruratanasirikul, Somchit Prince Songkla Univ, Pediatrics
REVIEW RETURNED	09-Dec-2021

GENERAL COMMENTS	The revised version of this manuscript is now much improved. There are still some minor mistakes on Page 8. 1. Individuals - A total 6883 women were recruited for a longitudinal study, 1875 individuals were excluded because the absence of metabolic measures, a group of 41 participants did not have medication history, and 65 individuals did not have their blood pressure, 389 individuals used antihypertension drugs. Finally, 4513 individuals were included in the analyses. - On Page 9, Anyone who appeared to unable to complete measurement in time due to health reasons such as hunchback and inability to stand was excluded from our study. Were these 'anyone 65 individuals did not have their blood pressure' included in the '65 individuals did not have their blood pressure'. This should be clear in the Methods section. 2. Measurement, Omron hem-7200 is the instrument for BP measurement, not height measurement. This instrument should be moved to For height measurement, should it be Stadiometer?
---

VERSION 2 – AUTHOR RESPONSE

Reviewer: 2

Dr. Somchit Jaruratanasirikul, Prince Songkla Univ

Comments to the Author:

The revised version of this manuscript is now much improved. There are still some minor mistakes on Page 8.

1. Individuals

- A total 6883 women were recruited for a longitudinal study, 1875 individuals were excluded because the absence of metabolic measures, a group of 41 participants did not have medication history, and 65 individuals did not have their blood pressure, 389 individuals used antihypertension drugs. Finally, 4513 individuals were included in the analyses.

- On Page 9, Anyone who appeared to unable to complete measurement in time due to health reasons such as hunchback and inability to stand was excluded from our study. Were these 'anyone 65 individuals did not have their blood pressure' included in the '65 individuals did not have their blood pressure'. This should be clear in the Methods section.

Response: thank you very much, we have revised the sample.

At baseline [22], 6883 women were recruited for a longitudinal study, 2370 individuals were excluded (i.e., absence of metabolic measures, and/or absence of medication history, and/or having no blood

pressure, and/or using antihypertension drugs). Finally, 4513 individuals were included in the analyses. Figure 1 summarized the selection of participants. Our study involved 4513 participants aged ≥ 45 years were total women [mean \pm standard deviation age= 58.59 ± 9.31 years, ranged from 45 to 96 years]. The mean and standard deviation of menarche age were 16.27 ± 2.18 years (ranged from 10 to 30 years).

2. Measurement,

Omron hem-7200 is the instrument for BP measurement, not height measurement. This instrument should be moved to For height measurement, should it be Stadiometer?

Response: thank you very much, we have revised the mistakes.

The average value of BP was measured by the mean of the 3-time measurements using Omron hem-7200 sphygmomanometer. Height was measured by height measurement instrument. Participants were asked to stand barefoot on the base plate of the height meter in the upright position, with the back against the vertical backplate of the measuring instrument, with eyes open, arms alongside the body and heels were closed and the toes were separated by 60 degrees. Anyone who appeared to be unable to complete measurement in time due to health reasons such as hunchback and inability to stand was excluded from our study. The method was used in our previous study [29].

Reviewer: 1

Dr. Basma Al-Ahmad, International Islamic University Malaysia

Comments to the Author:

the author already follow all the instructions and did all the requested changes that have been send to him previously, changes have been done accordingly

Response: thank you very much.

Reviewer: 2

Competing interests of Reviewer: None

Response: thank you very much.

Reviewer: 1

Competing interests of Reviewer: no competing interests and no area of conflict

Response: thank you very much.